# Duplex and Angiographic-Assisted Evaluation of Outcomes of Endovascular Embolization after Surgical Deep Vein Arterialization for the Treatment No-Option Critical Limb Ischemia Patients

**DOI:** 10.3390/diagnostics12122986

**Published:** 2022-11-29

**Authors:** Nunzio Montelione, Vincenzo Catanese, Teresa Gabellini, Francesco Alberto Codispoti, Antonio Nenna, Francesco Spinelli, Francesco Stilo

**Affiliations:** 1Division of Vascular Surgery, Department of Medicine and Surgery, University Hospital Foundation Campus Bio-Medico, 00128 Rome, Italy; 2Vascular Surgery Training School, Department of Translational Medicine and for Romagna, University of Ferrara, 44124 Ferrara, Italy; 3Cardiovascular Surgery, Department of Medicine and Surgery, University Hospital Foundation Campus Bio-Medico, 00128 Rome, Italy

**Keywords:** chronic limb-threatening ischemia, critical limb ischemia, no-option chronic limb-threatening ischemia, deep vein arterialization

## Abstract

Objective: To report early and mid-term outcomes of the arterialization of the deep venous system in no-option critical limb-threatening ischemia (CLTI) using duplex ultrasound and angiographic evaluation to improve limb perfusion. Methods: A single-center prospective study of patients with no-option CLTI treated with hybrid surgical arterialization of the deep venous circulation and staged endovascular embolization of the venous collateral. Embolization was performed using a controlled-release spiral, within two weeks after bypass surgery. Patients were assessed for clinical status, wound healing, median transcutaneous partial pressure of O_2_ (TcPO_2_), and post-operative duplex ultrasound evaluating peak systolic velocity (PSV), end diastolic velocity (EDV), and resistance index (RI) to assess foot perfusion and bypass features. Primary endpoint analysis was primary technical success, limb salvage, patency rates, and clinical improvement. Secondary endpoints were 30-day and long-term mortality, major cardiovascular events (MACE), including myocardial infarction or stroke, and serious adverse events (SAE). Results: Five patients with no-option CLTI were treated at our center using the hybrid deep vein arterialization technique. Clinical stage was grade 3 in one patient and grade 4 in the remaining four. Mean age was 65.8 years (range 49–76 years), and two patients were affected by Buerger’s disease. Primary technical success was achieved in all patients, and all the bypasses were patent at the angiographic examination. At 30-day and at average follow-up of 9.8 months (range 2–24 months), mortality, major cardiovascular events (MACE), and serious adverse events (SAE) were not reported, with a primary patency and limb salvage rates of 100%. Three patients required minor amputation. Clinical improvement was demonstrated in all patients with granulation, resolution of rest pain, or both. Median TcPO_2_ values rose from 10 mm Hg (range 4–25) before the procedure to 35 (range 31–57) after surgery, and to 59 mm Hg (range 50–76) after the staged endovascular procedure. Conclusions: In our initial experience, the arterialization of the deep venous circulation, with subsequent selective embolization of the venous escape routes from the foot, seems a feasible and effective solution for limb salvage in patients with no-option CLTI and those in the advanced wound, ischemia, and foot infection (WIfI) clinical stage.

## 1. Introduction

Chronic limb-threatening ischemia (CLTI) is the most severe clinical manifestation of peripheral arterial disease, and is characterized by ischemic pain at rest and/or tissue loss with ulceration and gangrene (Rutherford class 4–6), attributable to arterial occlusive disease [1].

According to the most recent classification, wound, ischemia, and foot infection (WIfI) scoring recognizes that a wide range of ischemic deficits may be limb-threatening when they coexists with varying degrees of wound complexity and superimposed infection [2].

In the United States and the European Union, more than 3.8 million patients suffer from CLTI [1], and this number is expected to increase by 23% over the next 10 years, in the context of an aging population with an increasing incidence of diabetes [3].

If CLTI is not promptly treated, it leads to a high risk of limb loss due to poor tissue perfusion [1].

Unsurprisingly, patients who present late and with the greatest degree of tissue loss are at highest risk, with reported rates of amputation at 4 years of 12.1%, 35.3%, and 67.3% for Rutherford class 4, class 5, and class 6, respectively [4]. Moreover, above-ankle amputation is associated with a cascading series of negative events that results in a mortality of 50% by 1 year and 75% by 5 years [5].

Unfortunately, not all CLTI patients can be revascularized according to standard surgical or endovascular techniques. Indeed, there are a small number of patients, defined as “no-option”, who present a lack of a target artery crossing the ankle and the absence of a suitable pedal or plantar artery target [6].

Surgical arterialization of the deep venous system, a technique that uses the venous bed as an alternative conduit for the perfusion of peripheral tissues with arterial blood through various techniques, has been performed since 1912 [7].

Recently, endovascular deep vein arterialization of the foot has been attempted in multiple settings using “off-label” [8] and purpose-built products [9,10] with a standardized approach to treating CLTI, to improve limb salvage and mortality in the no-option population.

The aim of the current work is to present our experience in the hybrid surgical and endovascular arterialization of the deep venous system of the foot for the treatment of patients with no-option CLTI, using duplex ultrasound and angiographic evaluation of outcomes.

## 2. Methods

A single-center prospective study of patients affected by no-option CLTI and electively treated by hybrid arterialization of the deep venous circulation, due to the lack of a suitable pedal or plantar artery target for surgical revascularization, was carried out between July 2020 and August 2022.

Written informed consent was given by all patients, and ethical approval for data analysis via a specifically created database was obtained by the institutional review board (approval number 31/20). This study was conducted in accordance with the Declaration of Helsinki.

Here, CLTI was classified according to the Rutherford and clinical stages of major limb amputation risk based on wound, ischemia, and foot infection (WIfI) classification [2].

The pre-operative assessment comprised clinical evaluation, ankle-brachial index (ABI), transcutaneous oximetry, and Doppler ultrasound examination. Selective computed tomography angiography or arterial angiography were performed pre-operatively (Figure 1A,B).

Transcutaneous partial pressure of O_2_ (TcPO_2_) at the level of the capillary bed was recorded with the TCM400 (Radiometer Medical ApS, Brønshøj, Denmark); each measurement lasted ~25 min, and they were taken before and after surgery.

Duplex ultrasound examination was performed using an ARIETTA V70, (Hitachi Ltd., Tokyo, Japan) with a 7 MHz linear probe at a 60° insonation angle to assess vein and artery features before surgery. After surgical and endovascular procedures, duplex examination peak systolic velocity (PSV), end diastolic velocity (EDV), and resistance index (RI) were calculated using the following formula: $\left(\left(peak\;systolic\;velocity-diastolic\;velocity\right)/\left(peak\;systolic\;velocity\right)\right)$, and were then evaluated at the terminal area of the foot, to assess the vascular supply to the foot.

During the remaining follow-up, graft patency was assessed with duplex ultrasound performed at 1, 3, and 6 months, and yearly thereafter.

## 3. Technical Notes

The hybrid approach consisted of a first bypass graft surgery to arterialize the deep venous circulation of the foot and a staged endovascular embolization of the retrograde venous drainage.

The “in situ” vein bypass using the great saphenous vein (GSV) was the preferred method for surgery. Veins were considered of good quality when they were at least 3 mm in diameter and did not present ectasia or post-phlebitic alterations. In cases where the great saphenous vein was not available or not long enough, alternative veins were considered.

The site of the proximal anastomosis was chosen according to pre-operative vessel patency assessment, and also confirmed by intraoperative evaluation. In the present series, a failed surgical exploration of the tibial and plantar arteries suggests the need for arterialization. Heparin was administered intravenously at a dose of 70 units/Kg before arterial clamping.

After the proximal anastomosis was performed, the valve lysis within the venous graft, was carried out using the Chevalier Valvulotome (LeMaitre Vascular, Inc., Burlington, MA, USA) even if the vein was inverted.

Then, the posterior perimalleolar tibial vein was isolated and clamped. After venotomy, the downstream valvular competence was unblocked by means of a coronary endoluminal probe (1.5–2 mm) to provide a low resistance of the arterial blood flow through the venous circulation of the foot.

The distal anastomosis was then performed at the level of tibial vein in a termino-lateral fashion with a continuous 7/0 polypropylene suture (Figure 2A). Then, the arterialized tibial vein is incompletely ligated using a 5/0 polypropylene distally to proximal anastomosis, partially reducing the blood theft from the foot. An intraoperative duplex scan examination was performed to assess the bypass’ patency.

Within two weeks after surgery, an angiographic examination was scheduled to confirm graft’ patency and to perform venous collateral embolization. In case of superficial femoral artery or popliteal proximal anastomosis, an antegrade percutaneous approach was performed. Alternatively, for a more proximal bypass (i.e., common femoral proximal anastomosis), a surgical exposure of the vein graft was performed usually at the middle portion of the thigh. After heparinization, over 0.014-inch Choice^TM^ guidewire (Boston Scientific Marlborough, MA, USA), with a 4Fr hydrophilic coated catheter Glidecath^®^ C1(Terumo, Somerset, NJ, USA), was advanced inside the vein graft, close to the distal anastomosis. Angiography was then performed to reveal the main venous collateral causing rapid wash-out of the contrast medium towards the leg, downstream of the distal anastomosis (Figure 2B,C). Then, after crossing the distal anastomosis, a selective catheterization of the collateral was achieved using the Progreat^®^ microcatheter system (Terumo, Somerset, NJ, USA), and embolization of the vessel was performed by positioning a controlled-release spiral (Penumbra, Inc., Alameda, CA, USA) to implement distal perfusion of the foot. (Figure 2D,G).

After surgery, acetylsalicylic acid (100 mg/die) and low-molecular-weight heparins (LMWHs) (70–100 UI/Kg twice a day) were given to patients. One month later, the LMWHs were suspended, followed by the provision of Rivaroxaban^®^ (2.5 mg twice a day).

Typical drawbacks of this type of surgery were the important stasis edema and the slow wound demarcation in case of foot gangrene. A “waiting strategy” was used to let the wound demarcate and to achieve wound healing, while a “tension-free” surgical approach was applied when minor amputation was needed, avoiding any primary intention wound closure. To reduce stasis edema, the patients rest with the limb unloaded throughout the entire first month.

## 4. Outcome Measures

The outcome measures were primary technical success, limb salvage, patency rates, and clinical improvement. Primary technical success was defined as successful surgical arterialization of the deep venous system and selective embolization of the main venous collateral. Clinical improvement was evaluated using granulation tissue formation and resolution of rest pain. Additionally increasing vascular perfusion of the foot was evaluated by means of TcPO_2_ and duplex examination.

Secondary endpoints were 30-day and long-term mortality, major cardiovascular events (MACE), including myocardial infarction or stroke, and serious adverse events (SAE), defined as respiratory failure, renal function impairment, major bleeding, graft infection, graft occlusion, and major amputation.

## 5. Results

Since July 2020, five CLTI patients were treated at our center using the hybrid deep vein arterialization technique. Demographic and clinical features of patient were described in Table 1. Pre-operative ABI was <0.4 in all patients. Two patients were classified as Rutherford 4, and three were Rutherford 5. The WIfI clinical stage was grade 3 in one patient and grade 4 in the remaining four. The mean age of patients was 65.8 years (range 49–76 years), with two patients affected by Buerger’s disease. In the present series, no patients showed chronic kidney disease.

Primary technical success was achieved in all patients. In four patients, the great saphenous vein was used as a conduit. In one patient, the small saphenous vein and the Giacomini vein were used (the great saphenous vein was already used in a previous surgery). All the bypasses were patent at the angiographic examination performed within two weeks after surgery. Selective embolization was achieved using a single controlled-release spiral in all patients, although in one a second spiral was required. Additionally, in one patient, an angioplasty of the valves of the deep venous circulation of the foot was performed during the endovascular procedure.

At 30-days and at the average follow-up of 9.8 months (range of 2–24 months), mortality, major cardiovascular events (MACE), and serious adverse events (SAE) were not reported, with a primary patency and limb salvage rates of 100%.

Three patients with pre-operative gangrene required minor amputations (Table 2). In two patients, an open trans-metatarsal amputation was performed (Figure 3), and in one patient, V finger amputation was performed.

Clinical improvement was demonstrated in all patients with granulation, resolution of rest pain, or both. There was a remarkable resolution of pain at rest in four patients immediately after surgery, as confirmed by the significant reduction in analgesic drugs in the post-operative period.

Although all patients rest with the limb unloaded, important stasis edema was reported in three patients, lasting five weeks.

Post-operative TcPO_2_ was increased in each patient compared to pre-operative values. Median values rose from 10 mm Hg (range 4–25) before the procedure to 35 (range 31–57) after surgery, and to 59 mm Hg (range 50–76) after the endovascular procedure.

The flow curves analysis performed during the ultrasound follow-up showed a systolic peak never exceeding 130 cm/s (mean PSV 76 cm/s) and a well-represented diastolic curve in all the patients (mean EDV 45 cm/s), typical characteristics of parenchymal destination flows (Figure 4). Duplex examination revealed a considerable increase in the mean RI after the endovascular embolization inside the bypass and at the level of the distal foot arteries (0.52 and 0.62 vs. 0.36 and 0.41, respectively).

## 6. Discussion

According to SVS guidelines, the definition of a no-option anatomic pattern of disease is dependent on clinical context; cases involving the lack of a target artery crossing the ankle and absence of a suitable pedal or plantar artery target may be considered no-option disease patterns in patients with advanced CLTI (e.g., WIfI clinical stages 3 and 4) [6].

All our patients fell into the category of no-option CLTI at an advanced WIfI clinical stage, in which traditional techniques did not lead to any therapeutic success, as demonstrated by previous unsuccessful attempts at endovascular or surgical revascularization performed in most of them, and especially for the cases of absent of adequate tibial or plantar arterials identified during the surgical exploration.

As a last resort, the arterialization of the deep venous system by making a vein-bypass in between the proximal arterial district and the deep venous circulation of the foot, creating an inverted arteriovenous flow, was performed to allow the perfusion of peripheral tissues with arterial blood.

The hybrid approach described in our study, combined previously described techniques [11,12,13,14].

Arterialization of the venous system by surgical bypass on the dorsal or plantar foot veins was applied for the first time by Lengua and Sheil in 1975 and 1977, respectively [11,12]. Since then, other surgical procedures were performed with promising results [13,14], also considering the high estimated lower limb amputation rate (10–40%) for untreated CLTI in the first six months and the associated poor survival rate (80% at 6 months, 50% at 5 years) [1,15,16]. The efficacy of surgical venous arterialization has been evaluated by a large meta-analysis involving 56 studies, which reported 71% limb salvage and 46% secondary patency at 12 months [17].

The surgical technique has evolved over time, as the first bypasses were performed at the level of the superficial venous circulation, causing severe edema and ischemia of the surgical wounds and a lack of perfusion of the foot.

Conversely, in the study of Mutirangura et al., unlike the approaches aimed at the superficial dorsal venous system, a surgical arteriovenous fistula of the plantar deep venous system using a composite bypass was proposed with a reported rescue limb rate of 76% at 24 months [18]. According to this study, due to the in-depth knowledge of the venous system, the deep venous circulation of the foot seems to be the best target for a higher tissue perfusion.

We believe that the choice of using a hybrid approach was decisive for the success of the revascularization in our series with 100% of primary patency and limb salvage rate at mean follow-up of 9.8 months; however, not all the patients have been reached during the 6-month follow-up at this time, and this is a possible limitation due to the previously reported high-risk of failure in the 6 months following the arterialization [14]. Another limitation of our study is the small sample size and the heterogeneity of the population comprising patients with Buerger’s disease and atherosclerotic disorder that are pathophysiology different.

A similar technique has been described by Alexandrescu [13], author of the first hybrid, angiosome-oriented approach. Because of the distal anastomosis of our bypass at the level of the posterior tibial vein, only ankle and foot veins were embolized, such as in Ferraresi’s study [14], with the aim to divert blood flow as distally as possible into the foot. For the first time, we proposed a well-defined and calculated timing for endovascular collateral embolization. Indeed, the secondary procedure of embolization was staged two weeks after the bypass surgery to reduce the risk of early vein graft occlusion due to an early increase in distal resistances. In addition, the maintenance procedures lead over time to a remodeling of the distal vascular network with an ever greater increase in peripheral perfusion.

To guide the correct timing of the secondary procedure, duplex ultrasound examination was performed with an accurate analysis of the flow curves. Peak systolic velocity, showing a systolic peak never exceeding 130 cm/s and a well-represented diastolic curve in all patients was visible, and these are typical characteristics of parenchymal destination flows. The mean value of post-embolization RI was 0.62, indicating how the arterialization of the deep venous system allows a retrograde perfusion of the microcirculation in a territory with terminal vascularization.

The exact mechanism of action for arterialization in the deep venous system is not yet clear. A theoretical explanation may be direct tissue nutrition provided by reverse perfusion through capillaries, collaterals, and arteriovenous shunts, and the stimulation of angiogenesis [19].

In all the studies of surgical or hybrid arterialization, except in the Ferraresi methodology [14], in which angioplasty was always performed to obtain the valve lysis, valvulotomy of the distal foot veins was by phlebotomy under direct visual control and/or retrograde passage of arterial dilators, irrigating cannulas, valvulotomes, cutting balloons, Fogarty catheters, or Parsonnet probes [11,12,13,14,18,19]. In our study, valve lysis was achieved by means of a coronary endoluminal probe, except in one patient, in which additional angioplasty was required during the endovascular embolization procedure due to a suboptimal surgical lysis. The correct destruction of the valves beyond the distal anastomosis is one of the crucial points of the surgical arterialization techniques. The valve lysis could be ameliorated with a new dedicated surgical antegrade valvulotome, such as the antegrade, over-the-wire valvulotome, recently proposed in the PROMISE I study. This trial represents an evolution of the first LimFlow experience [9,20], and also added self-expanding stent grafts specifically conceived to divert blood flow from the tibial artery into the tibial and pedal venous system with a reported procedural success of 75% and 30-day, 6-month, and 12-month amputation-free survival rates of 91%, 74%, and 70%, respectively [10].

However, secondary interventions in this patient population were still relatively high (52%). Most reinterventions were directed at the arterial inflow to the arterio-venous conduit due to the advanced and diffuse atherosclerotic disease affecting the inflow donor artery, and this could be a limit of the technique [10].

Despite the numerous positive experiences reported in the literature, there are conflicting opinions regarding the effectiveness of arterialization of the venous system. Indeed, as reported in Matkze’s study, 1-year limb salvage and survival rate in patients with CLTI treated with arterialization compared with patients CLTI-treated conservatively, patients were not significantly different (57% vs. 54% and 92% vs. 64%, respectively) [21]. The authors concluded that arterialization did not have any effect on limb salvage, and the results of the study did not support the use of arterialization as a treatment for CLTI [21]. However, considering the poor survival rate in the conservative group (64% vs. 92%), even if not statistically significant, it could be speculated that the unfavorable fate of non-revascularized patients could be related to the persisting of limb ischemia symptoms and the associated suffering.

Recently, the development of novel therapeutic angiogenesis cell sources, such as adipose-derived regenerative cells (ADRCs), has been investigated in no-option CLTI patients [22]. The results of this multicenter clinical trial demonstrated the safety, feasibility, and effectiveness of autologous ADRCs implantation for therapeutic angiogenesis, resulting in high major amputation-free survival rates (94.1%) at 6 months in no-option CLTI patients.

Further randomized, prospective studies with a larger number of patients, and comparing different technique of arterialization, therapeutic angiogenesis, and conservative treatment of no-option CLTI are required to validate the efficacy of arterialization.

## 7. Conclusions

In our initial experience, the arterialization of the deep venous circulation with subsequent selective embolization of the venous escape routes from the foot seems a feasible and effective solution for limb salvage in no-option patients at an advanced WIfI clinical stage.

Further and broader experiences with a longer follow-up are still needed to validate the clinical efficacy of the arterialization of the deep venous system in patients with no-option CLTI.

## Figures and Tables

**Figure 1 diagnostics-12-02986-f001:**
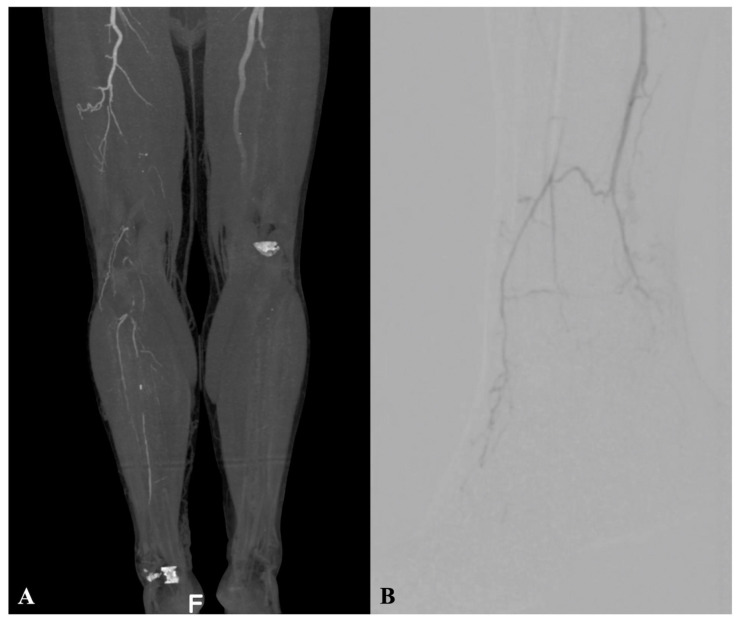
(**A**) Pre-operative computed tomography angiography showing posterior view of right limb distal SFA, with popliteal and infrapopliteal tibial vessels occlusion without patent vessels crossing the ankle in the patient with Rutherford 4 and CS 4; (**B**) baseline angiography in patient with advanced forefoot gangrene, exhibiting distal posterior tibial occlusion, partial recanalization of distal anterior tibial, and peroneal arteries without target artery crossing ankle into foot; P2 patterns according to inframalleolar/pedal disease descriptor in the Global Limb Anatomic Staging System (GLASS).

**Figure 2 diagnostics-12-02986-f002:**
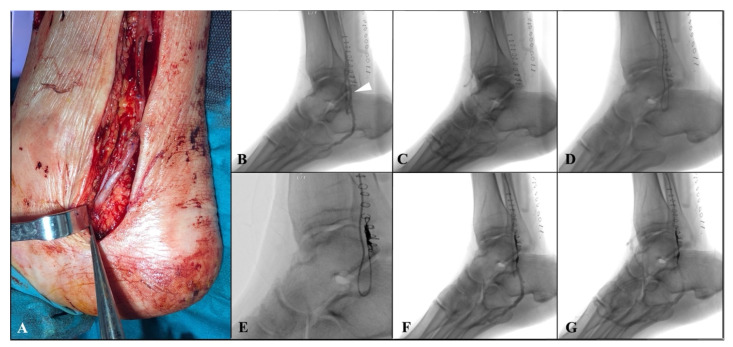
(**A**) Intraoperative details showing the distal anastomosis performed at the level of the perimalleolar tibial vein in a termino-lateral fashion; (**B**,**C**) angiographic examination, performed 10 days after first surgery, revealing bypass patency and the main venous collateral, causing rapid wash-out of the contrast medium towards the leg, downstream of the distal anastomosis (white arrow); (**D**,**E**) selective catheterization of the collateral and embolization by a controlled-release spiral; (**F**,**G**) final angiography showing the occlusion of the main venous collateral and the implemented distal perfusion of the foot.

**Figure 3 diagnostics-12-02986-f003:**
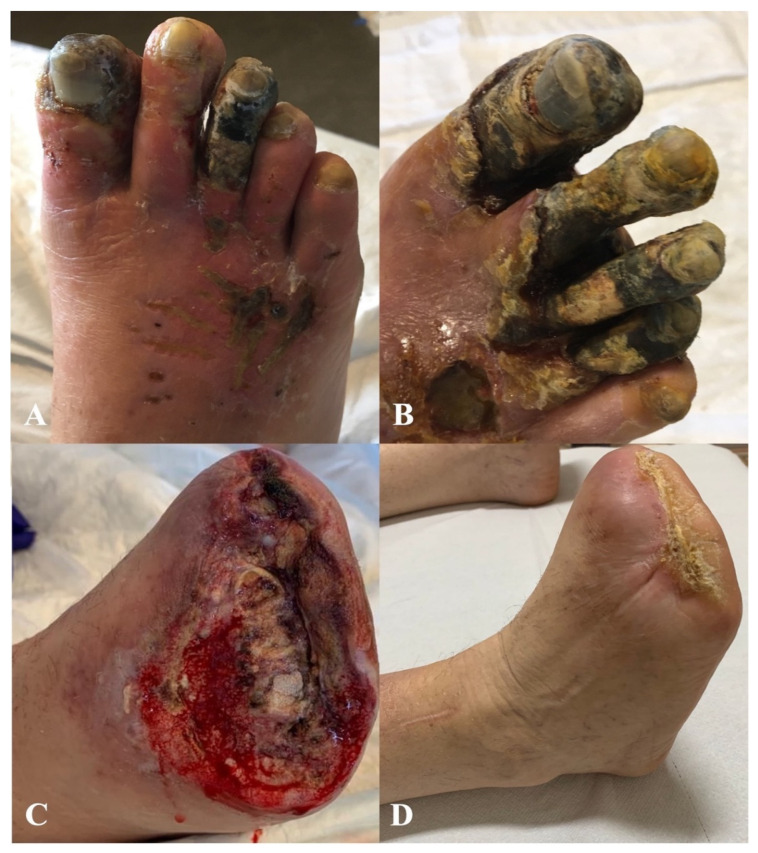
(**A**,**B**) Extensive forefoot gangrene; (**C**) open trans-metatarsal amputation performed after arterialization, avoiding primary closure according to the “tension-free” surgical approach; (**D**) complete healing of the amputation wound after VAC therapy.

**Figure 4 diagnostics-12-02986-f004:**
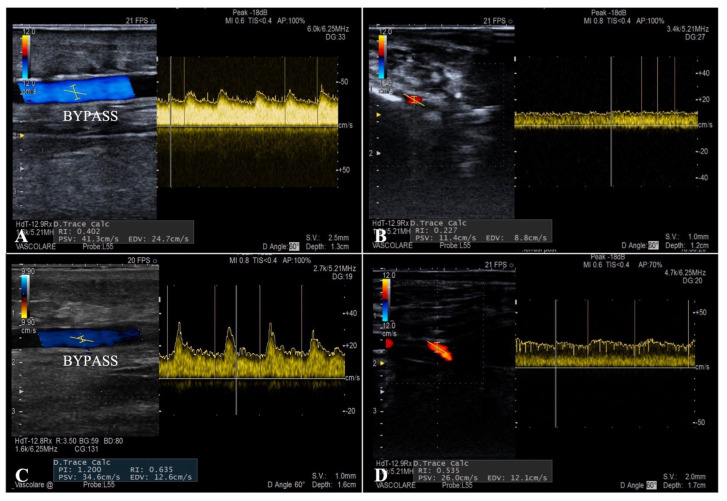
Post-operative duplex ultrasound examination showing bypass flow (**A**) and distal foot arteries perfusion (**B**) after surgical revascularization, and at the same levels after the embolization procedure (**C**,**D**).

**Table 1 diagnostics-12-02986-t001:** Pre-operative data. Demographics, risk factors, age, clinical features, and previous treatment.

Age, Years	65.8 ± 12.1
Men	4
Hypertension	4
Dyslipidemia	3
Diabetes mellitus	1
Smoker	3
CAD	1
COPD	1
Chronic kidney disease	0
ESRD/hemodialysis	0
Previous treatment	
Endovascular	2
Open surgery	1
Lesion characteristics (*n* = 5)	
WIfI wound grade	
0	2
1	0
2	1
3	2
WifI Clinical Stage	
1–2	0
3	1
4	4
Buerger’s disease	2
Location	
Superficial femoral artery	2
Popliteal artery	4
Tibioperoneal artery	4
BTK arteries	
0	0
1	0
2	1
3	4
Diffuse calcification	3

Abbreviations are as follows: BTK, below the knee; COPD, chronic obstructive pulmonary disease; ESRD, end-stage renal disease; WIfI, wound, ischemia, and foot infection. Continuous data are presented as the mean ± standard deviation; categorical data are given as the number.

**Table 2 diagnostics-12-02986-t002:** Intraoperative and post-operative details of enrolled patients.

Patient	BP Inflow	Graft	Major Amputation	Minor Amputation	Amputation Details
1	SFA	GSV in situ	No	Yes	Trans-metatarsal
2	CFA	GSV in situ	No	Yes	V finger
3	SFA	Composite: GSV+ superficial vein	No	No	/
4	SFA	SFV + Giacomini vein	No	Yes	Trans-metatarsal
5	Popliteal	GSV	No	No	/

Here, SFA = superficial femoral artery; GSV = great saphenous vein; CFA = common femoral artery.

## Data Availability

The data presented in this study are available on request from the corresponding author. The data are not publicly available due to privacy or ethical restrictions.

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
