# Peer review of "Duplex and Angiographic-Assisted Evaluation of Outcomes of Endovascular Embolization after Surgical Deep Vein Arterialization for the Treatment No-Option Critical Limb Ischemia Patients"

_diagnostics, 2022, doi:10.3390/diagnostics12122986_

Round 1
Reviewer 1 Report
The study `Duplex and Angiographic-Assisted Outcomes of Endovascular Embolization After Surgical Deep Vein Arterialization for the Treatment of no-option Critical limb Ischemia Patients´ aimed to present the author's experience in the hybrid surgical and endovascular revascularisation of the deep venous system for the treatment of patients with no-option CLTI. In this single-centre, prospective cohort study, five no-option CLTI patients were included. All patients received the arterialisation of the deep venous circulation with subsequent selective embolisation of the venous escape routes from the foot due to the lack of a target artery crossing the ankle and the absence of a suitable pedal or plantar artery target (according to SVS guidelines) in the clinical context of advanced CLTI. The primary endpoint was the primary technical success, limb salvage, patency rates and clinical improvement. After technically successful procedures in all patients, the authors report a primary patency and limb salvage of 100% at 30 days and an average follow-up of approximately ten months without major adverse cardiovascular events. They conclude that the hybrid technique seems a feasible and effective solution for limb salvage in these patients.
The effectiveness of distal venous arterialisation for salvage of critically ischemic limbs has been investigated in several studies with considerably diverging results. Nevertheless, the question is interesting in the real-world setting, especially since indications and technical procedures for revascularisation are heterogeneous. It seems all the more important to characterise the patients, their underlying disease leading to CLTI, and the clinical manifestation of CLTI, as well as previous revascularisation approaches to enable comparability and assessment of the transferability of the results.
Unfortunately, the manuscript shows numerous fundamental weaknesses, among those the following:
1. The fulfilment of the definition of no-option should be substantiated, and the localisation of the peripheral arterial occlusions should be described in more detail (also femoropopliteal, crural). A series of baseline angiograms (recommended for CLTI guideline adherence, see ref. 4) should be considered (in all five patients).
2. The tcpO2 measurement is undoubtedly necessary but also prone to error. It should be stated whether an ABI / TBI measurement was performed. Were the peripheral Doppler pressures all zero?
3. Where were the wounds located? All on the forefoot?
4. How was the clinical improvement assessed? Use of analgesics needed by patients with ischemic rest pain, but also postoperative patients?
5. The patient cohort was small (N=5) and heterogeneous: The two youngest patients with thrombangiitis obliterans and two others with diabetes mellitus are hardly comparable. The prognosis of the former depends mainly on smoking behaviour. At the same time, in the latter, it is not clear whether it is a diabetic foot syndrome in addition to peripheral arterial occlusive disease.
6. Rutherford 6 patients are considered severe CLTI in the literature with little or no hope of limb preservation. Please characterise the soft tissue lesions and whether the described malleolar surgery was possible.
7. Fig.3 A, C Angle of PW Doppler measurement not adequate for quantitative measurement of blood flow velocities. B, D has no acceptable longitudinal section and thus no good angle, so the flow velocities cannot be quantitatively evaluated and analysed. At the same time, the RI should be less affected here.
8. Several language corrections are needed, and abbreviations should be used consistently (CLTI vs CTLI; RI vs IR; WifI vs WIfI).
Author Response
- The fulfilment of the definition of no-option should be substantiated, and the localisation of the peripheral arterial occlusions should be described in more detail (also femoropopliteal, crural). A series of baseline angiograms (recommended for CLTI guideline adherence, see ref. 4) should be considered (in all five patients).
Thank you for the opportunity to elaborate on the thought process and methodology for evaluating preoperative assessment on this study. First, we would like the emphasize that this was a selective group analysis of patients with no-option CLTI in which arterialization of the deep vein circulation of the foot was performed as bail-out procedure. However, according to your valuable suggestion, the localisation of the peripheral arterial occlusions has been described in more detail, please see revised table 1.
While the authors understand that this is likely of great interest to readers, the bail-out nature of the procedure, precluded a standardized imaging protocol for this group of patients. Because of this, imaging that patient underwent preoperatively was quite heterogeneous. Also, is well reported in literature how angiography may occasionally fail to detect a patent distal artery target, and there are reports of successful tibial and pedal bypass grafting based on exploration of an artery identified on Doppler ultrasound examination that was not identified on contrast arteriography*,**.
Nevertheless, a baseline angiogram and a preoperative computed tomography angiography, performed in two of five patients, have been added in the revised manuscript, in order to clarify the arterial occlusions, please see picture 1 A-B.
*Pomposelli Jr FB, Jepsen SJ, Gibbons GW, Campbell DR, Freeman DV, Miller A, et al. Efficacy of the dorsal pedal bypass for limb salvage in diabetic patients: short-term observations. J Vasc Surg 1990;11:745e51. discussion: 751-2.
**Eiberg JP, Hansen MA, Jorgensen LG, Rasmussen JB, Jensen F, Schroeder TV. In-situ bypass surgery on arteriographically invisible vessels detected by Doppler-ultrasound for limb salvage. J Cardiovasc Surg (Torino) 2004;45:375e9
- The tcpO2 measurement is undoubtedly necessary but also prone to error. It should be stated whether an ABI / TBI measurement was performed. Were the peripheral Doppler pressures all zero? Thank you. We know that the tcpO2 measurement could be subject to errors. In all of our patients ABI measurement was performed preoperatively as first-line noninvasive test with suspected CLTI. This data has been added both in the methods and in the results section.
Peripheral doppler pressure at ankle level wasn’t always zero; indeed, in two patients were preoperatively detected monophasic flow pattern at duplex examination at the level of posterior and anterior tibial arteries. Unfortunately, in both cases the target artery was inadequate for distal anastomosis.
- Where were the wounds located? All on the forefoot? Thank you for giving us the opportunity the better clarify preoperative wounds condition. All the wounds were in the forefoot. A column was added in table I to have a clear visualization of wounds location. Please see also answer at point 6.
- How was the clinical improvement assessed? Use of analgesics needed by patients with ischemic rest pain, but also postoperative patients?
Thank you for the opportunity to clarify. We have added the following to clarify this:
“Remarkable was the resolution of pain at rest in 4 patients, immediately after surgery as confirmed by the significant reduction of analgesic drugs in the post-operative period”
- The patient cohort was small (N=5) and heterogeneous: The two youngest patients with thrombangiitis obliterans and two others with diabetes mellitus are hardly comparable. The prognosis of the former depends mainly on smoking behaviour. At the same time, in the latter, it is not clear whether it is a diabetic foot syndrome in addition to peripheral arterial occlusive disease.
Thank you for the opportunity to elaborate on the thought process and methodology for evaluating clinical outcomes in this heterogeneous study. First, we would like the emphasize that this was a subgroup analysis of cohort of patients affected by CLTI and submitted to arterialization of deep venous system as bail-out procedure. We know that patient cohort was small and heterogeneous; consequently, hardly comparable; however, this was not the focus of this study.
In present series, both diabetic patients showed a documented arterial occlusive disease as now reported in table 1. The two patients with thrombangiitis obliterans had stopped smoking for at least 3 years; also both patients underwent intravenous infusion of iloprost before considering the surgical approach and had previous treatments (one endovascular and one surgical) as shown in table 1.
- Rutherford 6 patients are considered severe CLTI in the literature with little or no hope of limb preservation. Please characterise the soft tissue lesions and whether the described malleolar surgery was possible. Thank you for the opportunity to clarify this point. Of course, Rutherford 6 patients showed major tissue loss above transmetatarsal level with little or no hope of limb preservation. However, none of the 5 patients in this series showed gangrene above the TM level and consequently could not be classified as Rutherford 6. It was a clear error by the authors when writing the manuscript. The manuscript has been corrected.
- Fig.3 A, C Angle of PW Doppler measurement not adequate for quantitative measurement of blood flow velocities. B, D has no acceptable longitudinal section and thus no good angle, so the flow velocities cannot be quantitatively evaluated and analysed. At the same time, the RI should be less affected here.
Thank you. As showed in picture PW angle was 60° in each evaluation and consequently ideal for blood flow assessment. B,D displayed distal foot arteries when a good longitudinal section in not possible.
- Several language corrections are needed, and abbreviations should be used consistently (CLTI vs CTLI; RI vs IR; WifI vs WIfI). Thank you. As per your suggestion, abbreviations were used consistently in the revised version of the manuscript. Language corrections have been made.
Reviewer 2 Report
In the study "Duplex and Angiographic-Assisted Outcomes of Endovascular Embolization After Surgical Deep Vein Arterialization for the Treatment of no-option Critical limb Ischemia Patients", the authors describe the mid-term outcomes of 5 patients treated since July 2020. All the bypasses were patent at the angiographic examination. Clinical improvement was demonstrated in all patients with granulation or resolution of rest pain. Secondary parameters like TcPO2 or resistance index improved as well. There occurred no MAZE or other serious adverse events. The authors concluded that the arterialization of the deep venous circulation with subsequent selective embolization of the venous escape routes from the foot, seems a feasible and effective solution for limb salvage in patients with no-option CLTI and advanced WIfI clinical stage.
Although only 5 patients were described in the study, the paper is an interesting contribution to the treatment of this rare non-option CLTI. However, there are some drawbacks that hinder from immediate publication:
But in detail:
Abstract: In addition to its information content, an abstract should also stimulate the interest of the readership through its writing style. Therefore, with almost 500 words, it is much too long and lacks stringency. It should therefore be revised and shortened to essentials.
Language: The English must be revised, particularly since beside grammatical errors also many spelling mistakes are contained, e.g. lines 61, 80-82, 86, 95, 182-185, 185, 202, 204-205, 225. Also judgements like "impressively" line 29 or 193, "dramatic" line 193, "indeed" line 204 or "Thanks..." line 257 should not be used, above all not in the abstract or result section. Furthermore, any abbreviation should be written out the first time it appears in the text, e.g. CLTI, WIfI or RI. Describing only 5 Patients, the % information, e.g. in the abstract or in lines 167-185, is unnecessary.
Introduction: acceptable
Methods: Overall, the procedure is well explained. However, a clearer line is needed here as well. For example, the "hybrid" approach should be shown more clearly - The word "hybrid" is not mentioned at all in the methods section. Redundancies should also be avoided to maintain stringency, such as, e.g. line 88-95 and line 154-155. On the other hand, line 86, ".. if required by senior author" could be more precise without unnecessarily lengthening the article. For the accuracy of the article, the beginning of the study (July 2020) as well as other information (number of patients, ethical committee vote, etc.) should already be mentioned in the methods and not only in the results section or even be missing.
Results: Especially here a more stringent presentation of the results would be desirable. A better description of the demographic data is needed. Table 1 is not sufficient. A summary table of the postoperative results is useful for clarification. By indicating the patient number, one can identify easily which of the patients needed "minor amputations" and which needed more significant surgery (metatarsal amputations). Furthermore, edema formation is one of the "typical drawbacks of this type of surgery" (line 146-147). An outcome in this regard does not appear in the results.
Figures: Please describe Fig 1 in legend in more detail. In which time period after surgery were the angiographies performed?
Discussion: Acceptable, but again, more stringency would be better. Comparison of own data to other methods comes late and should be more directly aligned with the cited studies. A (late) "returning to our series" is rather unfavorable.
Conclusions: acceptable
References: The literature is often incorrectly cited. Among other things, pages or years are missing. Also DOI is missing. Furthermore, there is newer literature that relates to the topic and therefore needs to be discussed.
Author Response
But in detail:
Abstract: In addition to its information content, an abstract should also stimulate the interest of the readership through its writing style. Therefore, with almost 500 words, it is much too long and lacks stringency. It should therefore be revised and shortened to essentials.
Thank you. Abstract has been shortened and revised according to your suggestion.
Language: The English must be revised, particularly since beside grammatical errors also many spelling mistakes are contained, e.g. lines 61, 80-82, 86, 95, 182-185, 185, 202, 204-205, 225. Also judgements like "impressively" line 29 or 193, "dramatic" line 193, "indeed" line 204 or "Thanks..." line 257 should not be used, above all not in the abstract or result section. Furthermore, any abbreviation should be written out the first time it appears in the text, e.g. CLTI, WIfI or RI. Describing only 5 Patients, the % information, e.g. in the abstract or in lines 167-185, is unnecessary.
According to your valuable suggestion, English has been revised and the spelling mistakes amended. All the abbreviation has been written out at first time it appears in the revised manuscript and % information has been removed due to the small sample size.
Introduction: acceptable
Methods: Overall, the procedure is well explained. However, a clearer line is needed here as well.
For example, the "hybrid" approach should be shown more clearly - The word "hybrid" is not mentioned at all in the methods section. Redundancies should also be avoided to maintain stringency, such as, e.g. line 88-95 and line 154-155. On the other hand, line 86, ".. if required by senior author" could be more precise without unnecessarily lengthening the article. For the accuracy of the article, the beginning of the study (July 2020) as well as other information (number of patients, ethical committee vote, etc.) should already be mentioned in the methods and not only in the results section or even be missing.
Thank you. The hybrid approach was defined at the beginning of the method section as per your suggestion. Most of the redundancies have been removed; however, considering the focus of the journal (Diagnostic), Authors prefer to preserving the detailed description in line 88-95 because essential for diagnostic assessment. Line 154-155 was shortened.
Line 86 were also well defined in the revised manuscript also to address advice coming from reviewer 1. Time of the study and others essential information (ethical committee…) were added in the method section of the revised manuscript.
Results: Especially here a more stringent presentation of the results would be desirable. A better description of the demographic data is needed. Table 1 is not sufficient. A summary table of the postoperative results is useful for clarification. By indicating the patient number, one can identify easily which of the patients needed "minor amputations" and which needed more significant surgery (metatarsal amputations). Furthermore, edema formation is one of the "typical drawbacks of this type of surgery" (line 146-147). An outcome in this regard does not appear in the results.
Thank you. Additional table with details of intraoperative details and postoperative results has been added in the revised manuscript (please see Table II). Edema information were added in the results section.
Figures: Please describe Fig 1 in legend in more detail. In which time period after surgery were the angiographies performed? A detailed description of figure 1 (Fig 2 in the revised manuscript) and time after bypass surgery was added.
Discussion: Acceptable, but again, more stringency would be better. Comparison of own data to other methods comes late and should be more directly aligned with the cited studies. A (late) "returning to our series" is rather unfavorable.
Thank you. The discussion section has been improved according to your suggestions
Conclusions: acceptable
References: The literature is often incorrectly cited. Among other things, pages or years are missing. Also DOI is missing. Furthermore, there is newer literature that relates to the topic and therefore needs to be discussed.
Thank you. Reference has been corrected; also DOI added.
Reviewer 3 Report
This report is of interest, though the material is small and procedures in Burger disease and atherosclerotic disorder are lumped together.
As reports on good outcome with conservative treatment of CTLI are published, it is of greater interest to compare such a management with deep vein arterialization (as in Mätzke´s Finnish study).
Further discussions on how to plan studies to search for evidence would be welcome
Author Response
As reports on good outcome with conservative treatment of CTLI are published, it is of greater interest to compare such a management with deep vein arterialization (as in Mätzke´s Finnish study). Thank you for suggestion. Matzke’s study has been added in the revised manuscript to consider conservative treatment of CLTI conversely to arterialization.
Further discussions on how to plan studies to search for evidence would be welcome.
Thank you. We have added the following to the discussion about furthers study: “ Further randomized prospective study, comparing different technique of arterialization and conservative treatment of no-option CLTI, with a larger number of patients, are required to validate the efficacy of arterialization”
Round 2
Reviewer 2 Report
Dear authors,
thank you for the revision of your manuscript. But there are still some aspects you need to work on:
-
In the affiliation "Vascular Surgery....", "Chief Prof. P. Zamboni" needs to be removed.
-
Language: Should be revised again. Beside typing errors, grammar and phrasing need to be checked, e.g. “…the present work is to present…” or “Table II”, “….using (ARIETTA V70, Hitachi Ltd, Tokyo, Japan)…”, no blanks e.g. (Fig.3), missing dots e.g. (Fig 3), “…a small of number of patients…” and so on.
-
Abstract: The abstract is now much better and more understandable. However, numbers below 10 should be written out, such as "remaining 4". Also the 60% should be taken out. The abbreviations SAE, MACE etc. are now written out in the text when they first appear, but WIfI again is not.
-
Introduction: acceptable, but the abbreviation POD is not necessary, since it does not appear in the text anymore.
-
Methods: The methods section is now better, but some revisions are required: The code number of the ethics votum is missing.
-
Results: How long did the patient have stasis edema? This is a common serious complication and affects the patients severely. Therefore, it is important to know how the patients tolerated it.
-
Especially since a relatively small case series is described here, the patient demographics and risk factors, preoperative data, etc. should be described in more detail. The information in Table 1 is not sufficient. How many patients had severe general atherosclerosis, insulin-dependent diabetes, smokers, CHD, renal disease, maxed-out medication, etc., etc.? As a template, the authors could use, for example, the work of Ferraresi doi.org/10.1177/1526602818820792.
-
Discussion: Acceptable, but again, comparison of own data to other methods comes late and should be more directly aligned with the cited studies.
-
References: Still, there is newer literature that relates to the topic and therefore needs to be discussed, e.g. doi.org/10.1007/s10456-022-09844-7. The study investigated safety, feasibility, and efficacy of therapeutic angiogenesis by cell transplantation. This other treatment should be discussed in the context to your therapeutic This other management of these patients, for example, should be discussed in the context of your therapy and outcome in the discussion section.
Author Response
Reviewer 2
thank you for the revision of your manuscript. But there are still some aspects you need to work on:
-In the affiliation "Vascular Surgery....", "Chief Prof. P. Zamboni" needs to be removed.
Thank you, the affiliation has been corrected.
-Language: Should be revised again. Beside typing errors, grammar and phrasing need to be checked, e.g. “…the present work is to present…” or “Table II”, “….using (ARIETTA V70, Hitachi Ltd, Tokyo, Japan)…”, no blanks e.g. (Fig.3), missing dots e.g. (Fig 3), “…a small of number of patients…” and so on.
Thank you, Language editing has been performed in the revised manuscript.
-Abstract: The abstract is now much better and more understandable. However, numbers below 10 should be written out, such as "remaining 4". Also the 60% should be taken out. The abbreviations SAE, MACE etc. are now written out in the text when they first appear, but WIfI again is not.
Thank you for suggestions; number has been written and also % removed. WIfi has been written out in text as is first time it appears.
-Introduction: acceptable, but the abbreviation POD is not necessary, since it does not appear in the text anymore.
Thank you, the abbreviation has been removed.
-Methods: The methods section is now better, but some revisions are required: The code number of the ethics votum is missing.
The code number of the ethics votum has been added
-Results: How long did the patient have stasis edema? This is a common serious complication and affects the patients severely. Therefore, it is important to know how the patients tolerated it.
Thank you. According to your suggestion, stasis edema details have been added in the result section of the revision manuscript.
-Especially since a relatively small case series is described here, the patient demographics and risk factors, preoperative data, etc. should be described in more detail. The information in Table 1 is not sufficient. How many patients had severe general atherosclerosis, insulin-dependent diabetes, smokers, CHD, renal disease, maxed-out medication, etc., etc.? As a template, the authors could use, for example, the work of Ferraresi doi.org/10.1177/1526602818820792.
Thank you. According to your suggestion, new Table I comprising more detailed description of clinical features of patients has been added in the revised manuscript.
-Discussion: Acceptable, but again, comparison of own data to other methods comes late and should be more directly aligned with the cited studies.
Thank you. According to your valuable suggestion, discussion has been revised
-References: Still, there is newer literature that relates to the topic and therefore needs to be discussed, e.g. doi.org/10.1007/s10456-022-09844-7. The study investigated safety, feasibility, and efficacy of therapeutic angiogenesis by cell transplantation. This other treatment should be discussed in the context to your therapeutic and outcome in the discussion section.
According to your suggestion, newer literature as the cited manuscript has been added and discussed in the discussion section.
Reviewer 3 Report
1. Although there is some interest in this presentation, it fails due to incorrect English. The authors are requested to be assisted by someone speaking English fluently, and who also understands the topic.
2. Title is not correct -outcomes can not be assisted by duplex etc. Suggest "Duplex ultrasound and angiographic evaluation of outcomes..........."
3. Abstract needs complete revision. There is a mix between Objective and Methods. CLTI should be explained the first time the abbreviation is used (Chronic Limb Threatening Ischemia). Details of duplex are not required in the abstract. Why delete tcpO2 (transcutaneous oximetry) ? Patient description is not part of Results. This subheading is mixing methods and results. There is no need to use percentages (20% vs 80%) for 5 subjects. An example of non-appropriate writing is "Impressively was the dramatic......".
Besides further need for English revision, Introduction covers the topic reasonably well. Methods - see comment on Abstract, but also "no-option CLTI" should stay. Patients are not "submitted". Why delete the paragraph on Ethical approval ?
Outcome measures are part of Results. Percentages are of limited interest, while it could be discussed if Buerger disease and atherosclerotic disorder should be lumped together. The two Buerger patients were -as expected- younger and pathophysiology is different.
Technical notes - The part starting "Inflow vessels..." is not understandable. As written one gets the impression that distal arteries were regarded "inflow".
The Discussion should preferably be shortened.
Author Response
- Although there is some interest in this presentation, it fails due to incorrect English. The authors are requested to be assisted by someone speaking English fluently, and who also understands the topic.
English has been revised along the entire manuscript, according to your suggestion.
- Title is not correct -outcomes can not be assisted by duplex etc. Suggest "Duplex ultrasound and angiographic evaluation of outcomes..........."
Thank you, Title has been changed according to your suggestion.
- Abstract needs complete revision. There is a mix between Objective and Methods. CLTI should be explained the first time the abbreviation is used (Chronic Limb Threatening Ischemia). Details of duplex are not required in the abstract. Why delete tcpO2 (transcutaneous oximetry)? Patient description is not part of Results. This subheading is mixing methods and results. There is no need to use percentages (20% vs 80%) for 5 subjects. An example of non-appropriate writing is "Impressively was the dramatic......".
Thank you, abstract has been revised.
- Besides further need for English revision, Introduction covers the topic reasonably well. Methods - see comment on Abstract, but also "no-option CLTI" should stay. Patients are not "submitted". Why delete the paragraph on Ethical approval?
Thank you, English has been revised and suggestions addressed.
- Outcome measures are part of Results. Percentages are of limited interest, while it could be discussed if Buerger disease and atherosclerotic disorder should be lumped together. The two Buerger patients were -as expected- younger and pathophysiology is different.
According to your suggestion percentages have been removed. Authors know that Buerger patients are pathophysiological different from atherosclerotic patients and this aspect has been added, as limitation of the study, in the discussion section.
- Technical notes - The part starting "Inflow vessels..." is not understandable. As written one gets the impression that distal arteries were regarded "inflow".
Thank you, the sentence has been corrected to avoid misunderstanding.
- The Discussion should preferably be shortened.
Thank you, Discussion section has been revised and shortened.
Round 3
Reviewer 2 Report
Dear authors,
thank you for the 2nd revision of your manuscript. But there are still some aspects you need to work on:
1. In the version "v3" that I can open, there is still the affiliation "Vascular Surgery...Chief Prof. P. Zamboni", although the authors intended to change the affiliation accordingly?!
2. Language: Should be revised again. In the version I have received, for example, the references are not superscript in the text. In addition, there are further incoherencies, e.g. Abstract: “…wound, ischemia, foot infection (WIfI) ….”, in the Introduction “….Wound, Ischemia, and foot Infection (WIfI) …”. The decision about a further professional language check must now be decided by the editorial office. Please explain: “Two patients were affected by Burger’s disease without patients with chronic kidney disease. (Line 171)”. I think a different formulation would be easier to understand.
Author Response
In the version "v3" that I can open, there is still the affiliation "Vascular Surgery...Chief Prof. P. Zamboni", although the authors intended to change the affiliation accordingly?!
Thank you, the affiliation has been changed according to your suggestion.
Language: Should be revised again. In the version I have received, for example, the references are not superscript in the text. In addition, there are further incoherencies, e.g. Abstract: “…wound, ischemia, foot infection (WIfI) ….”, in the Introduction “….Wound, Ischemia, and foot Infection (WIfI) …”. The decision about a further professional language check must now be decided by the editorial office. Please explain: “Two patients were affected by Burger’s disease without patients with chronic kidney disease. (Line 171)”. I think a different formulation would be easier to understand.
Thank you, references have been superscript in text when needed. Also WIfI definition and line 171 have been corrected in the revised manuscript, according to your suggestions.
Reviewer 3 Report
There is a concern that this minor material includes both Burger disease and atherosclerotic disorder but as a model for this kind of treatment it may be acceptable.
The title of the manuscript is still not adequate.
"Duplex and Angiographic-Assisted Outcomes of Endovascular Embolization After Surgical Deep Vein Arterialization for the Treatment of no-option Critical limb Ischemia Patients" .
Outcome is not caused by duplex etc. A reasonable modification would be "Duplex and Angiographic-Assisted evaluation of Outcomes of Endovascular Embolization After Surgical Deep Vein Arterialization for the Treatment of no-option Critical limb Ischemia Patients"
Author Response
There is a concern that this minor material includes both Burger disease and atherosclerotic disorder but as a model for this kind of treatment it may be acceptable.
The title of the manuscript is still not adequate.
"Duplex and Angiographic-Assisted Outcomes of Endovascular Embolization After Surgical Deep Vein Arterialization for the Treatment of no-option Critical limb Ischemia Patients" .
Outcome is not caused by duplex etc. A reasonable modification would be "Duplex and Angiographic-Assisted evaluation of Outcomes of Endovascular Embolization After Surgical Deep Vein Arterialization for the Treatment of no-option Critical limb Ischemia Patients"
Title has been modified according to your valuable suggestion. Please see the revised manuscrip